Intron gain by tandem genomic duplication: a novel case in a potato gene encoding RNA-dependent RNA polymerase

Ma Ming-Yue
Lan Xin-Ran
http://orcid.org/0000-0003-4503-2658 Niu Deng-Ke dkniu@bnu.edu.cn
MOE Key Laboratory for Biodiversity Science and Ecological Engineering and Beijing Key Laboratory of Gene Resource and Molecular Development, College of Life Sciences, Beijing Normal University , Beijing , China
Dessimoz Christophe
Electronic publication date: 2016 Jul 26
Publication date: 2016
Volume: 4
Electronic Location ID: e2272
Received 2015 Oct 16; Accepted 2016 Jun 29
Copyright: © 2016 Ma et al.
Copyright year: 2016
Copyright holder: Ma et al.
License: This is an open access article distributed under the terms of the Creative Commons Attribution License, which permits unrestricted use, distribution, reproduction and adaptation in any medium and for any purpose provided that it is properly attributed. For attribution, the original author(s), title, publication source (PeerJ) and either DOI or URL of the article must be cited.
License URL: https://creativecommons.org/licenses/by/4.0/

Keywords: Intron gain, Tandem genomic duplication, Splicing, RdRp, Solanum tuberosum, PGSC0003DMG402000361, Intron/exon structure

Funding: National Natural Science Foundation of China 31421063 and 31371283 This work was supported by the National Natural Science Foundation of China (grant numbers 31421063 and 31371283) and the Fundamental Research Funds for the Central Universities. The funders had no role in study design, data collection and analysis, decision to publish, or preparation of the manuscript.

==============================
The origin and subsequent accumulation of spliceosomal introns are prominent events in the evolution of eukaryotic gene structure. However, the mechanisms underlying intron gain remain unclear because there are few proven cases of recently gained introns. In an RNA-dependent RNA polymerase (RdRp) gene, we found that a tandem duplication occurred after the divergence of potato and its wild relatives among other Solanum plants. The duplicated sequence crosses the intron-exon boundary of the first intron and the second exon. A new intron was detected at this duplicated region, and it includes a small previously exonic segment of the upstream copy of the duplicated sequence and the intronic segment of the downstream copy of the duplicated sequence. The donor site of this new intron was directly obtained from the small previously exonic segment. Most of the splicing signals were inherited directly from the parental intron/exon structure, including a putative branch site, the polypyrimidine tract, the 3′ splicing site, two putative exonic splicing enhancers, and the GC contents differed between the intron and exon. In the widely cited model of intron gain by tandem genomic duplication, the duplication of an AGGT-containing exonic segment provides the GT and AG splicing sites for the new intron. Our results illustrate that the tandem duplication model of intron gain should be diverse in terms of obtaining the proper splicing signals.

Introduction

Although spliceosomal introns are the characteristic feature of eukaryotic nuclear genes, their origin and subsequent accumulation during evolution remain obscure. Several models of spliceosomal intron gain have been proposed, including intron transposition, transposon insertion, tandem genomic duplication, exogenous sequence insertion during double-strand-break repair, group II intron insertion, intron transfer, intronization and introner-like element insertion (van der Burgt et al., 2012; Yenerall & Zhou, 2012). Comparative analyses of discordant intron positions among conserved homologous genes have been conducted in diverse eukaryotic lineages. Although dozens of papers have reported intron gains over the last twenty years (Csuros, Rogozin & Koonin, 2011; Fablet et al., 2009; Hooks, Delneri & Griffiths-Jones, 2014; Irimia & Roy, 2014; Li et al., 2009; Li et al., 2014; Ma et al., 2015a; Roy & Gilbert, 2005; Roy & Penny, 2006; Torriani et al., 2011; van der Burgt et al., 2012; Verhelst, Van de Peer & Rouze, 2013; Yenerall, Krupa & Zhou, 2011; Yenerall & Zhou, 2012; Zhu & Niu, 2013a), only a few studies have identified the source sequences of these gained introns (Collemare et al., 2015; Denoeud et al., 2010; Hankeln et al., 1997; Simmons et al., 2015; Torriani et al., 2011; van der Burgt et al., 2012; Verhelst, Van de Peer & Rouze, 2013). In most studies (e.g., Knowles & McLysaght, 2006; Zhang, Yang & Niu, 2010), the source sequences have only been identified for a few introns, whereas tens of intron gains have been reported. Therefore, most of the reported intron gains do not provide supporting evidence for intron gain models. The mechanisms underlying these intron gains might be undetectable because the evolutionary traces have been erased by random mutations. Unexpectedly, the source sequences of most recent intron gains could not be identified. For example, researchers could identify the source sequences of only one intron gain among the seven new introns gained after the recent divergence (two Mya) of Drosophila persimilis and Drosophila pseudoobscura (Yenerall, Krupa & Zhou, 2011). More astonishingly, among the 21 new introns that were gained in certain local populations of Daphnia pulex, researchers successfully identified the source sequence of only one intron (Li et al., 2009). Because of the lack of identified source sequences, the mechanisms underlying most intron gains are not understood. Therefore, researchers have attempted to draw general conclusions from a small number of cases. Among the traditional models, intron gains by tandem genomic duplication should not occur at a low frequency because internal gene duplications are commonly observed (Gao & Lynch, 2009). This model was originally advanced by Rogers (1989), who suggested that the tandem duplication of an exonic segment harboring the AGGT sequence generates two splice sites for the new intron: 5′-GT and 3′-AG. In this model, a new intron is derived from the duplication of an exonic sequence, and the translated peptide is not altered by the intron gain. An example that is consistent with this model is the vertebrate gene ATP2A1 (Hellsten et al., 2011). The duplicated region of ATP2A1 has the AGGT signal and also includes a polypyrimidine tract and a branch point. In addition, the generation of the intron has been experimentally reproduced in a conserved paralogous gene, ATP2A2, by Hellsten et al. (2011). In fission yeasts, multiple tandem duplication of a 24 bp exonic segment containing AGGT has been observed in the genes SPOG_01682 and SOCG_00815. A comparison of these two genes with their expressed sequence tags indicates an intron across four duplicates in the gene SPOG_01682 and an intron across two duplicates in the gene SOCG_00815 (Zhu & Niu, 2013b). In the Arabidopsis TOUCH3 gene, Knowles & McLysaght (2006) observed two tandem internal gene duplications that duplicated an entire preexisting intron along with the exonic sequences on both sides of the intron. However, this finding does not represent the creation of new introns by tandem duplication but rather the multiplication of a preexisting intron by tandem duplication. Segmental duplication containing entire introns has also been observed by Gao & Lynch (2009). In the present paper, we confine our discussion of intron gain to the creation of new introns rather than the propagation of preexisting introns.

By comparing the orthologous genes of Solanum lycopersicum, Solanum tuberosum, and other Solanaceae plants, we found 11 cases of precise intron loss and six cases of imprecise intron loss (Ma et al., 2015b). Moreover, we found indications of an intron gain in one of the potato RNA-dependent RNA polymerase (RdRp) genes, PGSC0003DMG402000361 (Fig. 1). The RdRp genes encode enzymes that catalyze the replication of RNA from an RNA template, and these genes have been identified in all the major eukaryotic groups and play crucial roles in the regulation of development, maintenance of genome integrity, and defense against foreign nucleic acids (Willmann et al., 2011; Zong et al., 2009). In this study, we confirmed that the new intron was created by the duplication of a gene segment crossing one intron-exon boundary. The 5′ donor site of the new intron was activated by a cryptic donor site that previously occurred in the exonic region, whereas other splicing signals were inherited from the preexisting intron/exon structure.

Figure 1 Alignment of protein sequences close to the intron variation site.

The presence and absence of the intron are represented by 1 and 0, respectively. The genes shown in this figure are PGSC0003DMG402000361 in S. tuberosum, Solyc12g008410.1 in S. lycopersicum, Capana09g000243 in C. annuum, and Niben101Scf04189g00002 in N. benthamiana. The orthologous region in eggplant was manually identified by the reciprocal best BLAST hits and manually annotated. Abbreviations: Stub, S. tuberosum; Slyc, S. lycopersicum; Smel, S. melongena; Cann, C. annuum; and Nben, N. benthamiana.

Materials and Methods

The genome sequences and annotation files of domesticated potato S. tuberosum (PGSC_DM_v3), domesticated tomato S. lycopersicum (ITAG2.3), wild tobacco Nicotiana benthamiana (version 1.0.1), and wild tomato Solanum pennellii (spenn_v2.0) were downloaded from the Sol Genomics Network (Bombarely et al., 2011), and those for hot pepper Capsicum annuum L. (Zunla-1) were downloaded from the Pepper Genome Database (Qin et al., 2014). The scaffold sequences of Commerson’s wild potato (Solanum commersonii, JXZD00000000.1), another wild tomato (Solanum habrochaites, CBYS000000000.1), and eggplant (Solanum melongena, SME_r2.5.1) were downloaded from the NCBI Genome database (http://www.ncbi.nlm.nih.gov/genome/). The scaffold sequences and annotation files of Mimulus guttatus (version 2.0) were downloaded from Phytozome (https://phytozome.jgi.doe.gov/pz/portal.html). The following files were retrieved from the Sequence Read Archive of the NCBI (http://www.ncbi.nlm.nih.gov/sra/): SAR files of the whole-genome shotgun (WGS) reads (SRP007439) and the leaf, tuber, and mixed-tissue transcriptomes (SRP022916, SRP005965, SRP040682, and ERP003480) of S. tuberosum; the transcriptomes (SRP015739 and SRP018993) of S. lycopersicum; the transcriptome (SRP067562) of S. pennellii; the transcriptome (SRP019256) of C. annuum; and the transcriptome (SRP018508) of N. benthamiana. We mapped the RNA-Seq reads to the genomes using TopHat version 2.0.8 (Kim et al., 2013), whereas BWA (alignment via Burrows-Wheeler transformation, version 0.5.7) (Li & Durbin, 2009) was used for the WGS reads. We used the default parameters for both programs, although the minimum intron length was adjusted to 20 bp for TopHat.

The orthologous genes of PGSC0003DMG402000361 were identified using the best reciprocal BLAST hits with a threshold E value of < 10−10. In addition, the orthologous relationship between the genes in S. tuberosum and S. lycopersicum was confirmed by their synteny using SynMap (http://genomevolution.org/CoGe/SynMap.pl). Using the RNA-Seq data, we examined the available annotations of the RdRp genes in S. lycopersicum, S. pennellii, N. benthamiana, and C. annuum. The annotations in S. pennellii and N. benthamiana have been confirmed, and those in S. lycopersicum and C. annuum have been revised (Data S1). The orthologous sequences in S. commersonii, S. habrochaites, and S. melongena were manually annotated with references to the annotations in S. lycopersicum, C. annuum, and N. benthamiana. The annotation files are provided in Data S1.

According to the annotation files of the domesticated potato genome, PGSC_DM_v3, the 3′ end of the gene PGSC0003DMG402000361 overlaps with the 5′ end of the downstream gene PGSC0003DMG401000361 (Fig. S1). The orthologous genomic regions of C. annuum, M. guttatus, N. benthamiana, S. lycopersicum, and S. pennellii present a long gene sequence rather than two overlapping genes. We examined the annotation of this overlapping region using the RNA-Seq data of S. tuberosum and found a paired read (Read ID: 127022 in SRR866275) that crosses the overlapping region. It appears that a long transcript similar to the orthologs in other species occurs in S. tuberosum. In addition, the 3′ end of the coding sequence of the gene PGSC0003DMG402000361, GTAATCTGA, has been annotated as the beginning of the ninth intron of the longer transcript (Fig. S1). In total, we found 37 RNA-Seq reads that support the removal of the ninth intron from certain mature mRNA molecules and four RNA-Seq reads supporting the presence of the small segment GTAATCTGA in other mature mRNA molecules. In addition, we found transcription termination signals at the ends of both transcripts using POLYAH (Salamov & Solovyev, 1997). It appears that the potato RdRp gene PGSC0003DMG402000361 undergoes alternative cleavage and polyadenylation during transcription, which produces two isoforms with different lengths. We named the shorter one PGSC0003DMG402000361.S and the longer one PGSC0003DMG402000361.L. The newly identified intron is spliced from the identical region of these two transcripts; therefore, potential annotation errors in either transcript do not affect the validity of the identification of the new intron. For convenience, we present PGSC0003DMG402000361.L in this paper.

We found that the intron gain involved duplication using a BLAT search (Kent, 2002) and then identified the exact duplicated sequences using the programs REPuter (Kurtz et al., 2001) and Tandem Repeats Finder (Benson, 1999).

We searched the 5′ splicing sites, the branch sites, the polypyrimidine tracts, and the 3′ splicing sites according to Irimia & Roy (2008) and Schwartz et al. (2008). The exonic splicing enhancers (ESEs) of Arabidopsis thaliana were identified by Pertea, Mount & Salzberg (2007) and used as the query in a search of the 50 bp exonic sequences upstream and downstream of the target intron.

The phylogenetic tree was constructed using MEGA 6.0 by employing the maximum likelihood method with the Tamura-Nei substitution model and uniform rates (Tamura et al., 2013). The number of bootstrap replications was 1,000. The schematic diagram of the gene structures was constructed using the program GSDraw (Wang et al., 2013).

Results and Discussion

Among the cluster of orthologous genes for RdRp, the members of S. tuberosum and S. commersonii have 20 exons and those of the other Solanaceae species have 19 exons. A comparison of the annotations clearly showed that the second introns of S. tuberosum and S. commersonii are absent from the other Solanaceae genomes (Fig. 2). By analyzing the transcriptomic data of S. tuberosum, we found 106 RNA-Seq reads that were exclusively mapped to the annotated exon-exon boundary (Table S1; Fig. S2), which confirmed the annotation of this intron. Based on the phylogenetic tree constructed using the gene PGSC0003DMG402000361.L and its orthologs (Fig. 2), there were two possible explanations for the presence/absence of the intron: the gain of a new intron in the common ancestor of S. tuberosum and S. commersonii, or four independent intron loss events in the other four evolutionary branches (S. lycopersicum–S. habrochaites–S. pennellii; S. melongena; C. annuum; and N. benthamiana). According to the principle of parsimony, we concluded that the second intron of the gene PGSC0003DMG402000361.L was gained after the divergence of potato (S. tuberosum and S. commersonii) from other Solanum plants but prior to the divergence between S. tuberosum and S. commersonii.

Figure 2 Identification of the intron gain in potatoes.

The phylogenetic tree was constructed using the coding sequences of the gene PGSC0003DMG402000361.L and its orthologs Solyc12g008410.1 in S. lycopersicum, Sopen12g003370.2 in S. pennellii, Capana09g000243 in C. annuum, Niben101Scf04189g00002 in N. benthamiana, and Migut.K00531 in M. guttatus as well as the orthologous regions manually annotated in S. commersonii, S. habrochaites, and S. melongena. Numbers above the branches indicate the percentage of bootstrap support after 1,000 replicates. In the schematic diagram of the gene structures, the presented sequences start from the initiation codon ATG, the boxes represent exons, and the horizontal lines represent introns. Because of space limitations, the extraordinarily long introns are not scaled according to their lengths, and they are represented by broken lines. To avoid crowding together the slashed lines, the introns of M. guttatus have been scaled up by a factor of two. The new intron/exon structure is marked in red. Abbreviations: Stub, S. tuberosum; Scom, S. commersonii; Slyc, S. lycopersicum; Shab, S. habrochaites; Spen, S. pennellii; Smel, S. melongena; Cann, C. annuum; Nben, N. benthamiana; Mgut, M. guttatus.

The new intron and the inserted exonic sequence were used as a query sequence to search against the entire genome of S. tuberosum. We found that this insertion is a tandem genomic duplication (Fig. 2). The major part of the new intron and inserted exon region was a direct duplicate of the upstream intron-exon structure (Fig. 2). In addition, 10 nucleotides at the 5′ end of the new intron were recruited from the upstream exon (Fig. 2). Because two nearly identical regions in a reference genome might represent a true duplication or a false result caused by errors in genome assembly, we verified the duplication by examining the following three sources of evidence in S. tuberosum. First, 53 WGS reads were exclusively mapped crossing the three boundaries of two duplicates (Figs. S3–S5; Table S2). Second, 106 RNA-Seq reads were exclusively mapped crossing the mature mRNA exon boundary (Fig. S2; Table S1), which would not be observed in mature mRNA if the duplication had not occurred. Third, ten nucleotides were different between the duplicates (Fig. 3).

Figure 3 Splicing signals of the new intron in the potato gene PGSC0003DMG402000361.

Alignment of the two copies of the duplication. The splicing sites, the putative branch site, the polypyrimidine tract, and putative exonic splicing enhancers (TCAGCT, CAGCTC and GAGGAA) are underlined. A cryptic 5′ splicing signal, GTAAG, was activated by the duplication event. This duplication was also found in the orthologous region of the wild potato S. commersonii. In addition to this duplication, we detected another 83 bp tandem genomic duplication within the first intron of the gene PGSC0003DMG402000361 but not in the orthologous region of S. commersonii. The second duplication did not change the intron/exon structure of the gene PGSC0003DMG402000361; therefore, it is not described here in detail. Sites that differed between the two copies are indicated in green letters.

An intron is spliced out during the maturation of any RNA molecule, including protein-coding mRNAs and noncoding RNAs. In recent years, numerous spliced out sequences have been identified as originating in long noncoding RNAs and conclusively described as introns (Derrien et al., 2012; Guttman et al., 2009; Jayakodi et al., 2015; Kapusta & Feschotte, 2014). Therefore, the production of functional proteins by spliced RNA molecules should not be considered as a prerequisite for identifying a sequence spliced out of RNA molecules as a new intron. In this study, our search for evidence of intron gains is limited to whether the intron sequence occurred in the potato genome and whether the intron sequence has been removed from the mature mRNA. Although the WGS and RNA-Seq reads could demonstrate that the duplication is real and the intron is spliced, the best method of validating this assumption would be to perform a PCR assay for the genomic DNA and RT-PCR on RNA. This methodology underlies the objectives of our study.

According to Logsdon, Stoltzfus & Doolittle (1998), strong evidence of intron gain must satisfy two conditions. The first is a clear phylogeny that provides support for the intron gain, and the second is an identified source element for the gained intron. Because of the clear phylogeny and the identity of the source sequence, we consider the second intron of the potato gene PGSC0003DMG402000361.L to be a well-supported case of a newly gained intron.

According to the tandem genomic duplication model originally proposed by Rogers (1989), tandem duplication of an exonic segment harboring the AGGT sequence generates two splice sites for the new intron, 5′-GT and 3′-AG, and a new intron is derived from the duplication of the exonic sequence. However, the two splice sites do not contain sufficient information to unequivocally determine the exon-intron boundaries (Lim & Burge, 2001). Accurate recognition and efficient splicing of an intron also requires a polypyrimidine tract, an adenine nucleotide at the branch site, and many other cis-acting regulatory motifs (Schwartz et al., 2009; Spies et al., 2009; Wang & Burge, 2008; Wang et al., 2004). In addition, introns are often remarkably richer in A and U compared with exons (Amit et al., 2012), and this difference is considered a requirement for efficient splicing (Carle-Urioste, Brendel & Walbot, 1997; Luehrsen & Walbot, 1994). At first glance, it appears unlikely that a coding segment will have a full set of splicing signals. However, the intronization of coding regions has been observed in several different organisms, including animals and plants (Irimia et al., 2008; Kang et al., 2012; Szczesniak et al., 2011; Zhan et al., 2014; Zhu, Zhang & Long, 2009). These observations indicate that it is possible for certain coding sequences to contain a full set of cryptic splice signals. Furthermore, an experimental duplication of a coding segment of the vertebrate gene ATP2A2, which harbors the AGGT sequence, has been shown to generate the new intron (Hellsten et al., 2011). Therefore, a full set of the splicing signals required for active splicing is present in the coding sequence of the gene ATP2A2. Although a full set of the splicing signals may have been contained in the coding sequences, we believe that utilization of the active splicing signals of the parental intron/exon structure represent a more frequent method of intron gain. In the potato gene PGSC0003DMG402000361.L, the duplication includes the 3′ side sequence of an intron and the 5′ side of the downstream exon (Fig. 2). The 3′ splicing site signal (CAG), the polypyrimidine tract (TCTTCCAATGCCT), and the putative branch site (TTTAC) of this novel intron were inherited from the parental intron (Fig. 3). Moreover, the two overlapped putative ESEs of the 3′ flanking exon, TCAGCT and CAGCTC, and the GC contents that were different between the intron and exon (36% vs. 46%) inherited from the parental copy (Fig. 3). The 5′ splicing signal of the novel intron GTAAG was activated from a cryptic splice site that was recruited from the upstream exon. This case of intron gain indicates that the tandem duplication model should not be narrowly considered according to its original proposal 27 years ago (Rogers, 1989).

Conclusions

In the last common ancestor of domesticated potato S. tuberosum and wild potato S. commersonii, a tandem duplication event in the gene PGSC0003DMG402000361.L created a novel intron. The duplicate includes the 3′ side sequence of an intron and the 5′ side of the downstream exon. Most splicing signals that included a putative branch site, a polypyrimidine tract, a 3′ splicing site, two putative ESEs, and GC contents that were differentiated between the intron and exon inherited from the parental intron/exon structure. However, the widely cited model of intron gain includes the tandem duplication of an exonic segment containing AGGT, which would create the GT and AG splicing sites. The case of intron gain observed here illustrates that the tandem duplication model of intron gain should be diversified so that the proper splicing signals can be obtained.

Supplemental Information

Supplemental Information 1 Manually annotated RdRp genes in Solanum commersonii, Solanum habrochaites, and Solanum melongena.

Click here for additional data file.

Supplemental Information 2 Transcriptome data supporting the annotation of the target intron and genome sequencing data supporting the assembly of the region.

Table S1: The RNA-Seq reads mapped crossing the exon-exon boundary of the target intron; Table S2: The WGS reads mapped crossing the boundaries of the two duplicates.

Click here for additional data file.

Supplemental Information 3 Alignments showing the RNA-Seq reads and WGS reads mapped on the target gene.

Click here for additional data file.

We are thankful to the anonymous referees for their useful comments and Sidra Aslam for her help improving the English in this paper.

Additional Information and Declarations

Competing Interests

Author Contributions

Data Deposition

The authors declare that they have no competing interests.

Ming-Yue Ma analyzed the data, wrote the paper, prepared figures and/or tables, reviewed drafts of the paper.

Xin-Ran Lan analyzed the data, reviewed drafts of the paper.

Deng-Ke Niu conceived and designed the experiments, wrote the paper, prepared figures and/or tables, reviewed drafts of the paper.

The following information was supplied regarding data availability:

The raw data has been supplied as Supplemental Dataset Files.

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
