# Peer review of "Intron gain by tandem genomic duplication: a novel case in a potato gene encoding RNA-dependent RNA polymerase"

_PeerJ, doi:10.7717/peerj.2272_

## Round 0.1 · original submission · Major Revisions

The comments of the two reviewers are highly relevant. In particular, I agree with reviewer 1 that a functional study is required to support the notion that this is a bona-fide intron gain.

If you choose not to perform a functional assay, your findings would be considerably weaker. PeerJ's editorial policy sets no requirement in terms of significance, so we would still be interested in receiving a revised version. However, your manuscript would need to convey the tentative nature of your findings in the title, abstract, and main text.

Reviewer 1 ·

Basic reporting

The authors suggested a modification of an intron gain model using just one gene.

Experimental design

I would accept such attempt if the authors tried to validate that the resulting protein is really functional. Unfortunately the authors check a support from transcriptomic data without any functional analysis.

Validity of the findings

I think that the resulting protein is non-functional although a miricle cannot be excluded. In other words, this is a nonfunctional splicing isoform of the important gene (the RNA polymerase). The functional isoform have the same exon-intron structure as closely related orthologs.

Additional comments

The authors should do some more detailed functional studies if they want to insist on functionality of this new isoform. I think that people are not interested in nonfunctional products.

Reviewer 2 ·

Basic reporting

Ma and Niu identified an intron that was gained in an RdRp gene of potato by comparing the sequence of this gene to homologues in other Solanaceaeous plants. They found that this new intron is a partial copy of the upstream intron and exon that are conserved among Solanaceae. They propose to modify the mechanism of intron gain through genomic duplication.

Proven examples of how an intron has been gained are very rare and this finding should therefore be published. However, major modifications are needed before it can be accepted for publication.

1) The main modification is about the finding itself that is not fully correct. Indeed, the authors report the duplication of the 3’ end of the first intron and 5’ end of the second exon, plus a small insertion of an exogenous sequence of 10 bp. This is not correct because these 10 bp are found 100% conserved in the second exon. This means that the duplicated sequences has not been properly determined by the authors. They also should change their discussion because the donor site of the new intron comes from the duplicated exonic sequence.
In addition, when aligning homologues from potato and tomato, I realized that the first intron contains a duplicated sequence too, suggesting that this gene has experienced several tandem duplications. I assume the authors missed this duplication because they do not mention it. Did the authors check the whole gene for other tandem duplications?
The homologue in potato is shorter than the one in tomato. I think the authors need to present the whole gene annotation in the text and figures, describe accurately the gene models in all the species they used. By doing so, they can construct a real phylogenetic tree of this gene and compare this tree to a species tree in order to determine whether introns are conserved, gained or lost.

2) The introduction can be extended. The authors should cite Li et al., 2014 about intron gains in Daphnia. They should also better cite Hellsten et al. 2011, which remains the only experimental proof that genomic duplication can result in a new intron. In this study, the duplication involves an AGGT signal, but in addition polypyrimidin tract and branch point were present. The authors simplify the genomic duplication mechanism. I think it is clear that other signals than AGGT are needed. The examples of Hellsten et al., actually show that coding sequences may contain polypirimidin tracts and branch points. In the present study, the authors also show that potential donor sites can be present in coding sequences and their discussion should be modified accordingly. The authors do not have to argue for a different mechanism. It is the same as described before, but each example is different. In their case, the duplication led to a spliced intron thanks to the GTAAG sequence within the coding sequence. If the same duplication would have happen in tomato, it is unlikely the new intron would have been spliced because the sequence is ATAAG. Finding this intron is already enough and the authors should not try to oversell it with an argumentation that is not so convincing.

3) Is there any orthologue outside of the Solanaceae?
Figure 2: a real gene tree should be presented, and compared to the species tree (building a real one does not take so much time too)

4) I did not understand the purpose of measuring the dN/dS ratio. If the gene is expressed and introns are all properly spliced, the protein is likely to be functional. The amino acid insertion resulting from the duplication may have disrupted the function, but the only way to determine this is to introduce this sequence in an orhtologue or to revert the original sequence of the gene.

Figure 1 is not really clear. This way of presenting intron gain is usually within amino acid sequences. For nucleotide sequences, they can show the actual sequence.
Figure 3 is incorrect.
Table S1 and S2: a graphical representation would be better
line 18: «however» does not fit
line 63: I think it is not sensible to use top hat and BWA with default parameters and may lead to artefacts. Parameters of these tools have to be adjusted according to sequencing results.
line 67-70: this paragraph should be in the introduction or beginning of the results instead of methods.
line 78: what kind of information content can be calculated with WebLogo? This software provides a graphical representation of a consensus sequence based on an alignment.
line 86: what is an imprecise intron gain?

Experimental design

see basic reporting

Validity of the findings

see basic reporting

Additional comments

see basic reporting

---

## Round 0.2 · Major Revisions

In my previous decision, based on the referee's report, I asked you to either perform functional analyses to support your findings, or at least convey the tentative nature of some of the analysis.

However, as reviewer #2 reports, your revised manuscript remains overreaching in several respects, with basic checks lacking and unsubstantiated claims.

I am giving you another opportunity to revise, but please be aware that we will only do one more round of peer-review—if some of the points are not satisfactorily addressed, I will have to reject the submission.

Reviewer 2 ·

Basic reporting

The revised manuscript of Ma et al. properly addressed a few major issues (now they mentioned the origin of the acceptor site, figure 2 is clearer), but it still needs quite some improvement before it can be accepted. The authors are still trying to oversell their finding, which make them focusing on points or performing analyses that are not really relevant. Because of this, they do not perform basic checks and they miss interesting points to discuss.

The authors should provide the sequences they manually annotated. I could not find the NbS00003153g0003 gene at the Sol Genomics Network repository, is it the proper gene id? The new Figure 2 raises questions the authors should have addressed. It is striking that the gene in S. tuberosum is much shorter than the other homologues. The authors should have checked the annotation of this gene and would have realized that it is likely incorrect with the 9th intron not predicted, resulting in a shorter wrong gene model. The gene model in C. annuum looks also different, I would recommend the others to check this model too, as well as the one in N. benthamiana.
Another question is about the short extra exon predicted in Spen, Cann and Nben, was it deleted in other species, an insertion, a prediction problem? Although the authors focus on the upstream intron gain, this extra exon suggests an intron loss or gain during the evolutionary history of this gene and it is therefore interesting to mention if it is not a prediction problem.
I think it is always nicer to show the distance and bootstrap values in a phylogenetic tree. This tree should be rooted on the midpoint or the authors should use an outgroup sequence. The authors argue that they identified orthologues but they should provide a phylogenetic tree of homologues of PGSC0003DMG402000361, including more distant homologues. This would definitely prove their orthology. A quick blastp at ncbi gave many hits, so the authors should be able to provide a more extensive phylogeny with at least one orthologue that is not a Solanaceae. If sequences do not properly aligned at the nucleotide level, the authors can align protein sequences and also indicate in this protein alignment the position and phase of introns.

The authors mostly discuss the duplication they observe as a new version of the tandem genomic duplication mechanism for intron gain, but they do not discuss completely the duplication. Do the authors have an idea about the origin of the donor site? As mentioned in my previous review, the donor site is also duplicated from an exon. These two copies are found exactly at the borders of the duplicated sequence, which actually suggests that the donor site of the new intron might have been created thanks to DNA double strand break repair by homologous recombination. This should be discussed. Intron gain that involves such DNA repair was also found in other organisms.

I respectfully disagree with the authors that their finding is a new version of a model. I agree that the initial formulation of the model was too restrictive and I recommend the authors to present their work as an example of how diverse this model is in terms of getting proper splicing signals. As written in my previous review, I think it is nice to have another example that supports this mechanism of intron gain.

In their discussion, the authors suggest that tandem genomic duplications as an intron gain mechanism is underestimated and might be a major mechanism for intron gains in Eukaryotes, which has been overlooked. However, the argumentation about seeking in highly conserved orthologous genes does not prevent finding tandem genomic duplication as the authors identified such an event by comparing highly conserved orthologous genes. Thus, this methodology is unlikely to underestimate the number of intron gain by segmental duplications. I agree that this is a real mechanism that happens in all eukaryotes and is responsible for some intron gains, but the extent of these gains is still too limited to explain the origin and accumulation of spliceosomal introns in Eukaryotes. From their own results (Ma et al., 2015), they found 17 intron losses for only one intron gain.

I agree with the authors that the functionality of the protein does not matter in regards to intron gain and loss. Therefore, the whole discussion about protein functionality is not sensible and can be removed, including the non-sensible dN/dS analysis.

Following are additional comments:
The introduction is still very short and too simple.
Figure 1, Figure 2 and Figure 3 are redundant, they can be re-organized/combined (for example, figure 1 and figure 2 and figure 3a and 3b are very similar.
I do not really understand the purpose of Figure 3c and the analysis behind it because the new intron is supported by RNA-seq data, the donor and acceptor sites can be determined without knowing the consensus sequence based on all introns.
line 16: the identification of recently gained introns would be useful would read better
line 22-25: the wording suggests several duplication events. It should be clearer. One duplication event of an intron/exon junction, and the donor site of the intron likely originates from DSB repair.
line 86: TopHat parameters have to be adjusted according to the sequencing methods and results, for example the distance between mate pairs is important when mapping paired-end reads. I believe that the mapping was mostly accurate, but because we do not know the kind of reads that were mapped, it is difficult to say if these tools were accurately employed.
line 98-107: not clear, what are these 9883 groups of orthologous mRNAs? Was this done in this work or in the previous work published in Ma et al., 2015? If already published, the authors can just refer to that paper. If the aim of this search was to make a set of introns to determine the most frequent splicing signals in Solanum species, the authors did not need conserved introns only, they could have simply extracted intron sequences from the the three genomes using annotations.
line 109: which other parameters were used? how many bootstraps?
line 142: I acknowledge the effort to prove the duplication is real and the intron is spliced. The best way to prove this would have been to perform PCR on genomic DNA and RT-PCR on RNA. But I am convinced this duplication is real and the intron is spliced.

Experimental design

See basic reporting

Validity of the findings

See basic reporting

Additional comments

See basic reporting

---

## Round 0.3 · Major Revisions

As you can see, the remaining reviewer acknowledges progress but still noted quite a few points that need addressing. The reviewer has provided an extraordinarily detailed and constructive report, so I am hoping that this will help you solve the remaining points.

I'm tagging this round of revision as "major" because I view the requested points as compulsory, but I now feel that there is a real chance that we might be able to publish your study. Please address the reviewer's points in earnest in your revisions.

Please enlist the services of a professional English editor to address the outstanding language issues.

Reviewer 2 ·

Basic reporting

I appreciate the authors have addressed some of my comments, but there are very important ones that have not been properly addressed.

To be clear, accuracy of gene annotations is not my main interest, but it is a crucial point when studying intron gains and losses because results are affected by wrong annotations. The study of intron PAPs is about gene structure, therefore the authors should be concerned about the accuracy of annotations, otherwise it raises questions about the quality of their work. I initially made this comment because the wrong annotation of some of the orthologues was suggesting another intron or exon gain or loss in this gene and I wondered why the authors did not mention it. And in any case, it is important that everything in a study is accurate.
In regards to this exon, the authors have now modified the annotation of four genes (Scom, Slyc, Shab, Smel) in which the exon was missed by the automated gene predictors. This example should have convinced the authors that automated gene predictors often make mistakes and their predictions must be checked, and ideally confirmed by expression data. Of great importance, I disagree with the authors about the annotation of their gene of interest in S. tuberosum. My comment that the ninth intron present in all orthologues has been missed by the automated gene predictor is still valid for several reasons. The two genes predicted at the locus in Stub are both predicted as encoding RdRp proteins. I aligned the genomic sequence of this locus, covering both genes, to the cDNA of the Spen orthologue and it clearly shows that what the authors think is a stop codon is actually a trinucleotide within the missed intron and therefore has no effect on the protein sequence (see the files I made). When this ninth intron is properly predicted, the gene in Stub exhibits exactly the same structure as in Spen. In addition, the authors mention that they have RNA-seq support for this longer annotation, which is therefore real. Polyah provides a prediction only, I believe stronger in RNA-seq data. Altogether, it does appear that the gene predictor wrongly spit this gene in two. The legend of Figure 2 and Figure 2 are not correct.
In Spen, there are several annotations for this gene, and it seems that it could be a bit longer with 4 more exons, which would make the annotation of this gene the same as in Nben and Cann, which according to the authors are supported by RNA-seq data. I would recommend the authors to have another look at all these gene structures. I agree that it will not change their main result, but it makes the study of this gene inaccurate.

I still disagree with the last part of the discussion which attempts to state that intron gains by segmental duplication have been underestimated. The argumentation is quite confusing. The hypothesis made by the authors is based on other speculative hypotheses. First, the fact that a coding sequence has been gained alongside an intron is specific to this example and can be stand for segmental duplications, but it is not common to most observed intron gains. Then, such insertion will not result in a poorer alignment as the new sequence will not align. Poorly aligned regions are mostly due to high divergence time or fast evolution rate. This is the reason why it is important to discard positions that are poorly aligned. I do not understand the argument of evolutionary distances between organisms because it is normally possible to find species that have not diverged so long ago, which allows the identification of intron gains and losses. Using such reliable datasets, it appears that segmental duplication is not the major mechanism for intron gains. Even in their previous paper, the authors agree that the "imprecise" intron losses they observed could be due to independent events, which is particularly true for the first example they give in this paper. A ratio of 2 out of 443, and more likely 1 to 443, is not frequent to me. And "imprecise" intron gain has not been observed in this previous dataset. The frequency of segmental duplications is not that high and I do not think it can be stated that such events represent an important force that shaped gene structure.


The English writing has improved, although errors and gramatically incorrect sentences remain. As I am not co-author and not native speaker, I think it is inappropriate to ask me about any wording. I guess PeerJ will be of help when it comes to make a proof of the article.


Other minor points:
- Title should not contain abbreviations
- The abstract is not easy to read. For example, the second sentence or the "upstream copy" and "downstream copy" are difficult to understand. The origin of the donor site should be mentioned. The sentence about the other intronic duplication is not required in the abstract.
- l. 47-51: this is not exactly true because models are based on observations. In the case of ILEs, the source sequences are known and in the tunicate Oikopleura, near identical introns support the transposition mechanism. There are also examples that support intron transfer between paralogues.
- l. 59: what is more important about this model is that it is the only one that was experimentally proven.
- How the phylogenetic tree was built is not detailed enough. Which substitution model was used, which rates, etc?
- l. 139: absent also in the outgroup, important to mention.
- l. 146-147: actually three loss events.
- l. 153-155: the whole insertion is a direct duplication, not a major part.
- l. 168-172: what do the authors mean?
- l. 175: knowing the source of the intron gain is not a recognized, because not needed, criterium to assign an intron gain.
- l. 197: "efficient" is not the adequate word, it is likely more frequent.
- l. 207-209: this is too speculative because the authors did not show that the GAGGAA sequence is an ESE, just a prediction. According to this sentence, any GAGGAA sequence in the genome is a potential ESE.
- Figure 2A and B could be fused as it was in the previous version, in MEGA you can change the width of the tree.

Experimental design

see basic reporting

Validity of the findings

see basic reporting

Additional comments

see basic reporting

---

## Round 0.4 · accepted · Accept

Thank you for thoroughly addressing the outstanding comments, and editing the manuscript for English. The paper is now acceptable for publication. Congratulations!